# The Scania Accelerated Intermittent Theta-burst Implementation Study (SATIS)–Lessons from an accelerated treatment protocol

**Marcus Persson**[1]*, **Viktor Fabri**[1,2], **Alexander Reijbrandt**[1,3], **Annika Lexén**[4], **Hans Eriksson**[1,5], **Pouya Movahed Rad**[6]

1 University Psychiatric Department, Helsingborg, Sweden, 2 Department of Adult Psychiatry, Malmö, Sweden, 3 Hadsibrandt AB, Helsingborg, Sweden, 4 Department of Health Sciences/Mental Health, Activity and Participation, Lund University, Lund, Sweden, 5 HMNC Brain Health, Munich, Germany, 6 Department of Clinical Sciences, Faculty of Medicine, Lund University, Lund, Sweden

* marcus.persson@med.lu.se

## Abstract

### Background and objective

The Scania Accelerated Intermittent Theta-burst Implementation Study (SATIS) aimed to investigate the tolerability, preliminary effectiveness, and practical feasibility of an accelerated intermittent theta burst stimulation (aTBS) protocol in treating depression.

### Methods

We used an open-label observational design, recruiting 20 patients (aged 19–84 years) from two public brain stimulation centers in Sweden. During the five-day treatment period and at a follow-up visit after 30 days we closely monitored adverse events and collected self-rated side effect data. Objective (MADRS, CGI) and subjective (MADRS-S) measures of symptoms and functioning (EQ-5D) were also assessed. Feasibility was evaluated using direct patient ratings combined with a qualitative approach evaluating staff experience.

### Results

All patients reported adverse events at some point, the most common being headache (18/20 patients), but they were generally transient. MADRS scores decreased from 28.4 (min = 17, max = 38. SD = 6.9) at baseline to 20.0 (min = 1, max = 42. SD = 11.6) after the last day of treatment. 25% (n = 5) met the response criteria, with a mean time to response of 2.2 days (min = 1, max = 3. SD = 1.1). The practical arrangements surrounding this new treatment proved challenging for the organization, but patients reported few practical problems.

### Conclusion

SATIS provided further insights into the potential benefits and challenges associated with aTBS protocols. Effectiveness and drop-out rates were comparable to national data of conventional iTBS, but with a markedly faster time to response. More resources were required than anticipated, increasing the clinical workload.

**Data Availability Statement:** All relevant data are within the manuscript and its Supporting Information files.

**Funding:** Data was collected as a clinical project at the University Psychiatric Department in Helsingborg - a part of Region Skåne. Costs for APC and ethical review were covered by grants from the private foundation "Stiftelsen Söderström-Königska sjukhemmet" (grant-number SLS-570781) (PM-R). The private foundation "Stig och Ragna Gorthons stiftelse" provided financial support for MP (grant number 2023-2963) to analyse data and write the paper. No sponsor were directly involved in any way in the conduct of the research or preparation of the article. https://www.sls.se/vetenskap/sok-anslag/stift.-soderstrom/ https://www.gorthonstiftelsen.se/.

**Competing interests:** The authors, Marcus Persson, Viktor Fabri, Alexander Reijbrandt, Annika Lexén, and Pouya Movahed Rad, have no competing interests to declare. Hans Eriksson is currently a full-time employee at HMNC Brain Health, a pharmaceutical company developing treatments for depression. This does not alter our adherence to PLOS ONE policies on data and material sharing.

## Introduction

Depression is a prevalent and debilitating mental health condition, affecting approximately 4.7% of the global population [1]. Beyond profound psychological suffering, depression increases the risk of physical morbidity and premature death, significantly contributing to disability-adjusted life years [2].

Current antidepressant therapies yield unsatisfactory responses in up to 30% of patients and require weeks to manifest their full effects [3]. Although electroconvulsive therapy (ECT) is the primary treatment option for severe depression, it requires anesthesia and is associated with cognitive side effects that persist in roughly 17% of patients six months after treatment [4]. Hence, there is a pressing need for novel antidepressant treatments that provide rapid, lasting relief, while being well-tolerated and safe.

One such innovative approach is repetitive transcranial magnetic stimulation (rTMS), a method for noninvasive brain stimulation that has demonstrated effectiveness in the treatment of depression [5]. rTMS involves the use of an electromagnetic coil placed on the scalp to modulate electrical signaling in the cortex, with repeated sessions leading to changes in functional connectivity [6]. Treatment is typically given once every weekday for 4 to 6 weeks.

A further development in rTMS treatment; intermittent theta burst stimulation (iTBS), has expanded treatment capabilities by drastically reducing session duration from 37 to 3 minutes while maintaining efficacy comparable to standard rTMS [7]. In Sweden iTBS has become the predominant form of rTMS with a response rate of 42% and remission rate of 16% according to national registry data [8].

Building on these advances, Cole et al [9] introduced an accelerated iTBS protocol (aTBS) termed SAINT, which involves spaced delivery of stimulation for 10 daily sessions of iTBS with a 50-minute intersessional interval over 5 consecutive days, totaling 90,000 TMS pulses compared with 12,000 pulses in a standard iTBS protocol [7]. This novel approach also incorporates functional connectivity magnetic resonance imaging (fcMRI) for precise individualized targeting. Remarkably, 19 out of 21 patients met remission criteria after treatment, with 30% achieving remission within the first day of treatment. These findings highlight aTBS as a promising treatment protocol for depression offering practical advantages when compared to standard rTMS approaches.

In this study we aimed to assess the tolerability, preliminary effectiveness and practical feasibility of implementing a modified aTBS-protocol in a clinical real-world setting using an open-label observational design.

## Methods

### Participants and eligibility

We recruited 20 consecutive patients between August 15, 2021 and November 10, 2021, from the two public centers of brain stimulation in the Scania region (population 1.36 million) of Sweden, located in Lund and Helsingborg. These centers receive rTMS-referrals from primary care units, outpatient mental health services in the region, and local inpatient mental health services.

Patients included in this study were required to have moderate to severe depressive symptoms, either as part of a major depressive disorder (296.22, 296.23, 296.24, 296.32, 296.33, 296.34), a bipolar disorder (296.52, 296.53, 296.54), or a persistent depressive disorder (300.4) as defined by DSM-5 criteria [10].

At the screening stage, patients needed to be of at least 18 years of age, have a Montgomery-Åsberg Depression Rating Scale (MADRS) [11] score of ≥20, a negative urine drug screen,

phosphatidylethanol in blood (B-Peth) ≤0.05 µmol/L and a negative urine pregnancy test if female of reproductive age.

Patients were excluded if their symptoms were better explained by another diagnosis (i.e., a personality disorder, cognitive impairment or a neuropsychiatric disorder), if they had clinical contraindications to rTMS (i.e. metal implants in the head/neck region) or were undergoing compulsory care. Patients with a history of seizures, a family history of epilepsy, or other relevant neurological conditions were also excluded due to an increased risk of seizures during treatment [12]. See Fig 1.

Nurses at the Helsingborg brain stimulation center, where all treatments were administered, were invited to participate in semi-structured interviews before and after the active treatment phase of the study.

## Data collection

We developed two semi-structured interview guides for the nurse interviews; one for use before the treatment phase of the trial and one for use after. Both guides included seven principal questions (expectations, worries, hopes, workload, work environment, practical problems and resource effectiveness).

With the semi-structured format providing a framework, the interviewer (HE) used open questions to encourage participants to elaborate. Both pre- and post-treatment phase interviews lasted approximately 20 minutes and were digitally recorded. All five nurses (female) completed the interviews.

The nurses were also asked to rate their expectations regarding treatment effect using a Visual Analog Scale (VAS) (-5 = no effect, 0 = as effective as the standard FDA-approved iTBS-protocol, +5 = all participants in remission after treatment) and side effects pretreatment using VAS (-5 = extremely serious side effects, 0 = side effects on par with the standard FDA-approved iTBS-protocol, +5 = no side effects). After all treatments were administered, the nurses were asked to estimate treatment effect and side effects using the same scales again.

## Clinical assessments

In Table 1 we summarize the demographic information, treatment history, MADRS, clinical global impression (CGI) [13] and EQ-5D (EuroQol-5 Dimension incl VAS) [14] scores, as well as current diagnosis of the participants at the screening visit. All participants were required to maintain their ongoing antidepressant regimen during the treatment phase (see S1 Table), but during follow-up all interventions were allowed, including ECT.

We administered MADRS-S (the self-assessment version of MADRS) and EQ-5D at baseline and after each day of treatment. After the first treatment session and before each following session we asked the participants about any adverse events (AEs). We specifically queried about known rTMS-associated side effects, and asked participants to describe other perceived side effects. Before the first and after the 5th, 19th and 33rd treatment, we administered MADRS, the Comprehensive Psychopathological Rating Scale memory item (CPRS-M) [15], CGI- severity (CGI-S) and CGI—improvement (CGI-I). At the same checkpoints we evaluated tolerability and effectiveness using VAS (tolerability; 0 = worst possible side effects, 10 = no side effects at all. Effectiveness; 0 = no treatment effect at all, 10 = I have been cured). To evaluate practical feasibility, we used a VAS (practicality; 0 = practically impossible to go through with treatment, 10 = no practical problems) after the 5th, 19th and 33rd treatment. At a follow-up visit 30 days after the treatment we evaluated patient experience including the VAS assessments, AEs and administered MADRS, MADRS-S, CPRS-M, EQ-5D, CGI-S and CGI-I.

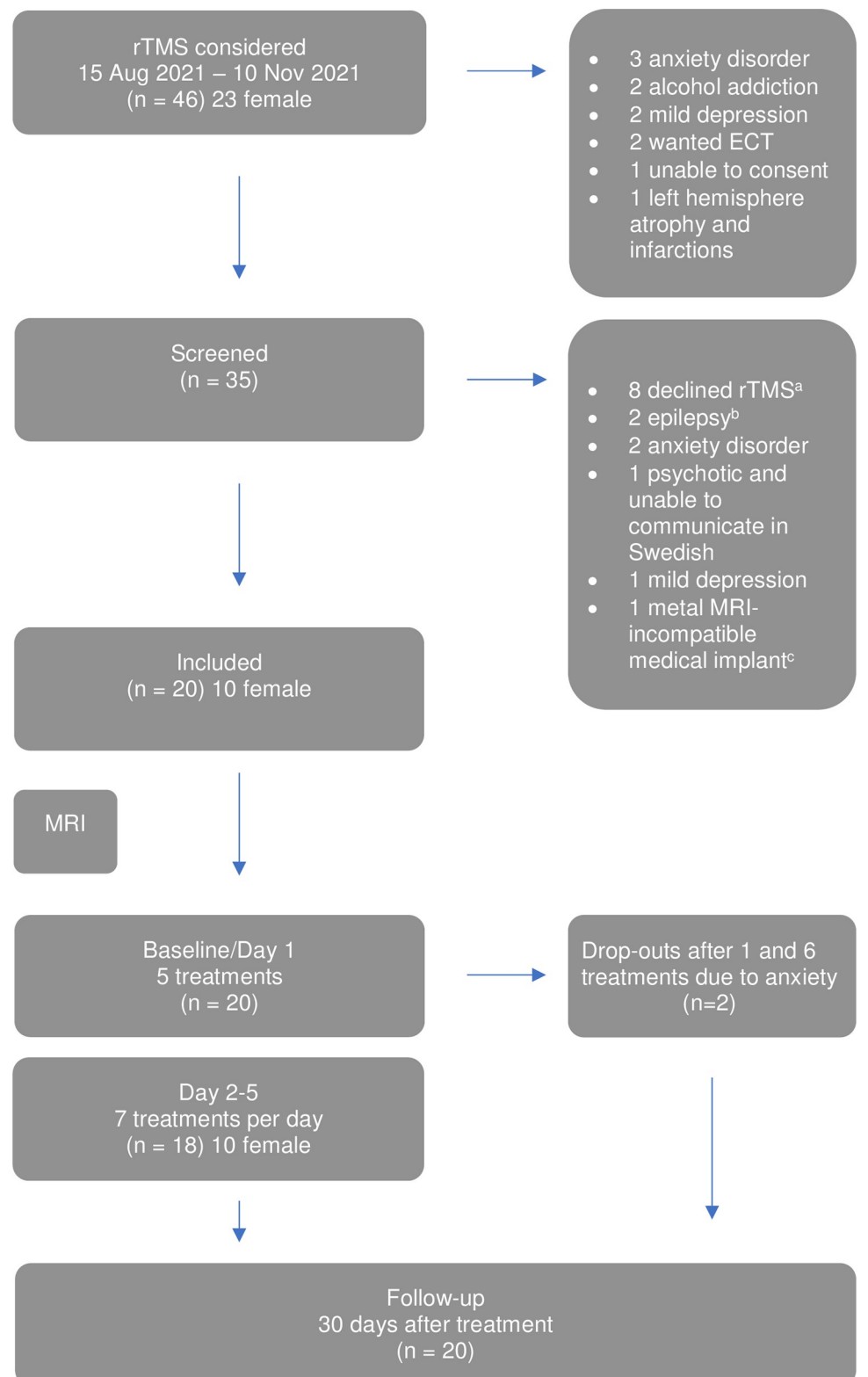

**Fig 1. The Scania aTBS implementation study (SATIS) trial profile.** rTMS repetitive Transcranial Magnetic Stimulation, ECT Electroconvulsive Therapy, MRI Magnetic Resonance Imaging. [a] Wanted therapy instead, wanted ECT instead, unable to come to the clinic because of agoraphobia, did not want to participate, unable to travel to Helsingborg for treatment, got diagnosed with autism the day before and needed time to adjust, unable to attend

treatment until early 2022, did not want to stop medication with benzodiazepines and antiepileptics. [b] Sibling with epilepsy, suspected epileptic seizure requiring electroencephalogram before treatment which was not possible to obtain prior to end of inclusion period. [c] Dorsal column stimulator from 2002.

**Table 1. Demographic information and treatment history for all participants (N = 20) in the Scania aTBS implementation study (SATIS) at inclusion.** ECT Electroconvulsive therapy, rTMS repetitive transcranial magnetic stimulation, MADRS Montgomery Åsberg Depression Rating Scale, EQ-5D EuroQol-5 Dimension, VAS Visual Analogue Scale, CGI Clinical Global Impression.

| Characteristic or Measure | Mean | SD | Min-Max |
|---|---|---|---|
| Age (years) | 45 | 19.99 | 19–84 |
| Duration of current depressive episode (years) | 3.23 | 4.92 | 0.12–20 |
| Age at onset of first depressive episode (years) | 28.8 | 18.57 | 12–83 |
| Number of depressive episodes including current episode[a] | 5.15 | 5.55 | 1–25 |
| Number of depressive episodes requiring inpatient care (lifetime)[b] | 1.95 | 2.61 | 0–8 |
| Number of medication trials current episode | 3.25 | 2.12 | 1–9 |
| MADRS | 30.95 | 6.78 | 21–48 |
| CGI-S (N = 16) | 4.19 | 0.91 | 3–6 |
| EQ-5D-3L mobility (N = 18) | 0.11 | 0.32 | 0–1 |
| EQ-5D-3L self-care (N = 18) | 0.17 | 0.38 | 0–1 |
| EQ-5D-3L usual activities (N = 18) | 1.33 | 0.69 | 0–2 |
| EQ-5D-3L pain/discomfort (N = 17) | 0.65 | 0.79 | 0–2 |
| EQ-5D-3L anxiety/depression (N = 18) | 1.72 | 0.57 | 0–2 |
| EQ-5D-3L VAS (N = 18) | 18.72 | 13.29 | 0–55 |
| | **n** | **%** | **% of treated** |
| Female | 10 | 50 | 55.6 |
| Major Depressive Disorder: Single Episode, Moderate (296.22)[c] | 1 | 5 | 5.6 |
| Major Depressive Disorder: Recurrent Episode, Moderate (296.32) | 8 | 40 | 38.9 |
| Major Depressive Disorder: Recurrent Episode, Severe (296.33) | 2 | 10 | 11.1 |
| Major Depressive Disorder: Recurrent Episode, With Psychotic Features (296.34) | 1 | 5 | 5.6 |
| Bipolar I Disorder, Most Recent Episode Depressed, Severe Without Psychotic Features (296.53) | 1 | 5 | 5.6 |
| Bipolar II Disorder, Most Recent Episode Depressed, Moderate/Severe (296.89) | 2 | 10 | 11.1 |
| Persistent Depressive Disorder With Persistent Major Depressive Episode, Moderate/Severe (300.4) | 5 | 25 | 22.2 |
| Had at least one comorbid psychiatric condition (Autism, Personality disorders, Anxiety disorders, etc) | 12 | 60 | 61.1 |
| Participants who attempted ECT (lifetime) | 8 | 40 | 38.9 |
| Participants who attempted rTMS (lifetime)[d] | 1 | 5 | 5.6 |
| Referrals from primary care units | 1 | 5 | 5.6 |
| Referrals from secondary outpatient mental health services[e] | 14 | 70 | 66.7 |
| Referrals from inpatient mental health services | 5 | 25 | 27.8 |

[a] 25% of the participants were experiencing their first depression or had never remitted from their depressive state.

[b] Several participants had been admitted repeatedly during the same episode, but we only registered one inpatient stay per episode. 25% of the patients had never been hospitalized.

[c] Diagnoses from DSM-5 Diagnostic and Statistical Manual of mental disorders, 2013.

[d] tried rTMS twice, the first time with partial response, the second time with no response.

[e] 5/14 in this group came from Lund, all other participants were from the Helsingborg area.

If a patient did not meet the criteria for remission at the follow-up visit, we offered a standard FDA-approved 4-week iTBS treatment.

## Intervention

We utilized the Nexstim NBT 2 (Nexstim OY, Helsinki Finland) system to deliver 60 cycles of 10 bursts of three pulses at 50Hz in two second trains (5Hz) with an 8 second intertrain interval. Five iTBS sessions were given the first day, followed by 7 daily sessions over four consecutive days. The first day included only five treatments due to the additional time needed for initial patient preparation, motor threshold determination and equipment setup. Treatments were administered by registered nurses, with a physician (MP) participating during start-ups and some sessions.

1800 pulses at 120% of the motor threshold were delivered to the left dorsolateral prefrontal cortex (DLPFC) per session, with an intersessional interval of 50 minutes and a 110-minute lunch break between the fourth and fifth session each day.

Studying individual MRI-scans from each patient, we located the DLPFC using neuroanatomical landmarks as described by Mylius et al [16]. We used MRI guided neuronavigation (Nexstim NBT 2) ensuring consistent delivery to the treatment target. 17 out of 20 patients required initial titration of the treatment dose. See Table 2 for a comparison between the standard high-frequency rTMS (HF-rTMS), the FDA-approved iTBS, SAINT and SATIS. Between treatments participants were seated in a common waiting room or laying down in the ECT recovery ward. We scheduled treatments every 20 minutes letting us treat a maximum of three participants per week.

## Ethical considerations

The study was approved by the Swedish ethical review authority (Dnr 2021–03290). All participants (patients and nurses) gave their written informed consent before participating in any study procedures.

**Table 2. A comparison between different rTMS protocols.** HF-rTMS high-frequency repetitive transcranial magnetic stimulation, iTBS intermittent theta burst stimulation, SAINT Stanford Accelerated Intelligent Neuromodulation Therapy, SATIS Scania aTBS implementation study.

| Parameter | HF-rTMS | iTBS | SAINT | SATIS |
|---|---|---|---|---|
| Intensity (% of motor threshold) | 120 | 120 | 90* | 120 |
| Frequency (Hertz) | 10 | 50/5 | 50/5 | 50/5 |
| Pulse train duration (seconds) | 4 | 2 | 2 | 2 |
| Intertrain interval (seconds) | 26 | 8 | 8 | 8 |
| Session duration (minutes, seconds) | 37,30 | 03,17 | 09,51 | 09,51 |
| Intersession interval (hours, minutes) | 24,00 | 24,00 | 00,50 | 00,50 |
| Pulses per session | 3000 | 600 | 1800 | 1800 |
| Sessions per day | 1 | 1 | 10 | 7† |
| Total number of sessions | 30 | 20 | 50 | 33 |
| Total treatment duration (days) | 42 | 28 | 5 | 5 |
| Total pulses | 90000 | 12000 | 90000 | 59400 |
| Method of neuronavigation | - | - | rs-fMRI** | MRI |

* Depth correction to consistently achieve 90% of resting motor threshold at the depth of the functional target, max 120% of resting motor threshold.

† Only five sessions day 1.

** Resting-state functional MRI.

## Data analysis

We conducted all statistical analyses using Excel (Microsoft® Excel® for Microsoft 365 MSO (Version 2208 Build 16.0.15601.20676)). In case of missing data, the last observation was carried forward. The level of statistical significance was set at p = 0.05. Analysis was carried out for all participants receiving at least one treatment session (n = 20) on an intention to treat (ITT) basis, as well as based on the group that completed the treatment program per protocol (PP) (n = 18).

Our primary outcome measures included adverse events during and after treatment, change in MADRS-score from baseline to immediately after the last treatment session and qualitative content analysis of staff experience with the implementation. Response was defined as a ≥50% reduction in the MADRS score, and remission as a MADRS-score of <11.

Secondary outcome measures included nurse-rated side effect load (VAS), patient-rated side effect load (VAS), patient-rated practicality (VAS), daily MADRS-S and EQ-5D-scores, CGI-I after the last treatment and at the 30-day follow-up.

We conducted qualitative content analysis on the interviews [17]. The recordings were transcribed verbatim. Two of the authors (VF,MP) read the transcriptions of the interviews repeatedly to gain a comprehensive understanding of the whole. Initial thoughts and reflections were discussed among the authors (VF,MP).

Each meaningful information unit from the interviews was then condensed and assigned a code. The codes were compared based on similarities and differences, sorted into subcategories and then grouped into main categories. The first interview yielded four main categories and the second interview yielded six. To ensure rigor, an author with extensive experience in qualitative research (AL) reviewed the interviews and the analysis.

## Results

### Tolerability

During the study, two serious adverse events (SAE) were reported. One participant attempted suicide two days after completing the treatment, while another reported persistent headache and a return of "buzzing in the head" at the 30-day follow-up. Two patients (aged 45 and 27) discontinued treatment due to intolerable anxiety, one after the first session, and one after six sessions.

At the 30-day follow-up, only six patients reported AEs. However, all patients reported adverse events at some point when queried after each session. Headaches were most frequently reported, with 18 out of 20 participants experiencing them during at least one treatment session, and 13 reporting headaches between sessions. Muscle twitching and anxiety between sessions were each reported by 16 patients. Additionally, 12 reported anxiety during a treatment session. Five patients experienced dizziness at least once during treatment, and ten reported dizziness between sessions. AEs reported by fewer than ten patients are detailed in S2 Table. No patient experienced seizures or syncope during the study.

We used VAS scores to assess anticipated side effects (10 indicating no side effects at all). Pretreatment ratings at group level were 7.5 (min = 3, max = 10. SD = 2.3) both in the ITT analysis and the PP-group. After the last treatment, patients rated their side effects at an average of 7.6 (min = 1, max = 10. SD 2.7 and 2.8 respectively) in both groups. At the 30-day follow-up, side effects were rated at 8.4 in the ITT analysis (min = 0, max = 10. SD = 2.9) and 8.7 in the PP-group (min = 2, max = 10. SD 2.2).

Subjective memory functioning measured as mean CPRS-M score in the PP-group (n = 18) was 2.2 (min = 0, max = 4. SD 1.7) before the first treatment, 1.6 (min = 0, max = 4. SD 1.7) after the last treatment and 1.7 (min = 0, max = 4. SD 1.7) at the 30-day follow-up. Self-rated

CPRS-M in the same group was 2.4 (min = 0, max = 5. SD 1.7) before the first treatment, 2.1 (min = 0, max = 4. SD 1.8) after the last treatment and 1.7 (min = 0, max = 5. SD 1.7) at the 30-day follow-up.

## Preliminary effectiveness

In the PP-group (n = 18), the mean MADRS score decreased from 28.6 (min = 17, max = 38. SD = 6.9) at baseline to 19.3 (min = 1, max = 42. SD = 11.8) after the last day of treatment as illustrated in Fig 2. This reduction in MADRS score represents an average reduction of 31.8%. A paired T-test comparing the MADRS score in this group at baseline and after the last day of treatment reveals a significant reduction in depression score with a mean difference of 9.2 points (SD = 11.2, p = 0.003). In this analysis 28% (n = 5) met response criteria and 17% (n = 3) also achieved remission. The mean and minimum MADRS values at baseline were affected by a drop in scores between inclusion and the start of treatment.

In the ITT analysis (n = 20) the mean MADRS score decreased from 28.4 (min = 17, max = 38. SD = 6.9) at baseline to 20.0 (min = 1, max = 42. SD = 11.7) after the last day of treatment. At that time 25% (n = 5) of patients met the response criteria and 15% (n = 3) met the remission criteria (see S1 Fig).

Mean time to remission was 3 days (min = 1, max = 5. SD = 2). Mean time to response was 2.2 days (min = 1, max = 3. SD = 1.1).

At the 30-day follow-up the mean MADRS score was 19.2 (min = 2, max = 38. SD = 11.4) in the PP group and 19.7 (min = 2, max = 38. SD = 11) in the ITT analysis. Three patients (17%) received ECT between the last day of treatment and the follow-up visit. Excluding these patients from the PP-group (n = 15), the mean MADRS score was 21 (min = 3, max = 38. SD = 11.2) and four patients (27%) met remission criteria (see S2 Fig).

As expected, CGI-S scores reflected the improvement in MADRS score. The greatest improvement was observed between baseline and after the first day of treatment. The mean CGI-S score dropped from 4.17 (min = 3, max = 6. SD = 0.92) before treatment to 3.33 (min = 1, max = 6. SD = 1.53) after the last treatment in the PP-group (n = 18). In the ITT analysis (n = 20) mean CGI-S score dropped from 4.15 (min = 3, max = 6. SD = 0.93) at baseline to 3.4 (min = 1, max = 6. SD = 1.5) after the last day of treatment. CGI-I scores changed notably between end of treatment and the 30-day follow-up (see Fig 3).

EQ-5D-VAS scores reflected the MADRS-S scores with an even rate of improvement during treatments shown in S3 Fig. In the PP-group the mean EQ-5D VAS score rose from 27.5 before treatments (min = 6, max = 52. SD = 15.27) to 41.5 (min = 7, max = 90. SD = 25.61) after 33 treatments. Functioning measured in the domains of mobility, self-care, usual activities, pain/discomfort and anxiety/depression also showed improvement.

As rTMS was first introduced in 2019 as a publicly available treatment in the region, only one participant had previous experience with it. This patient did not meet the criteria for response after treatment.

A correlation analysis conducted on the data in the PP-group revealed a Pearson correlation coefficient of 0.39 between the patients' pretreatment VAS-rated expectations for treatment effect and the subsequent change in their MADRS scores (see S4 Fig). This suggests a low positive correlation, indicating that higher expectations (VAS scores) were associated with greater improvements in depression symptoms.

## Practical feasibility

The content of the semi-structured interviews with the nurses who were involved in performing the treatment are presented in Tables 3 and 4. Interviews conducted before the treatment

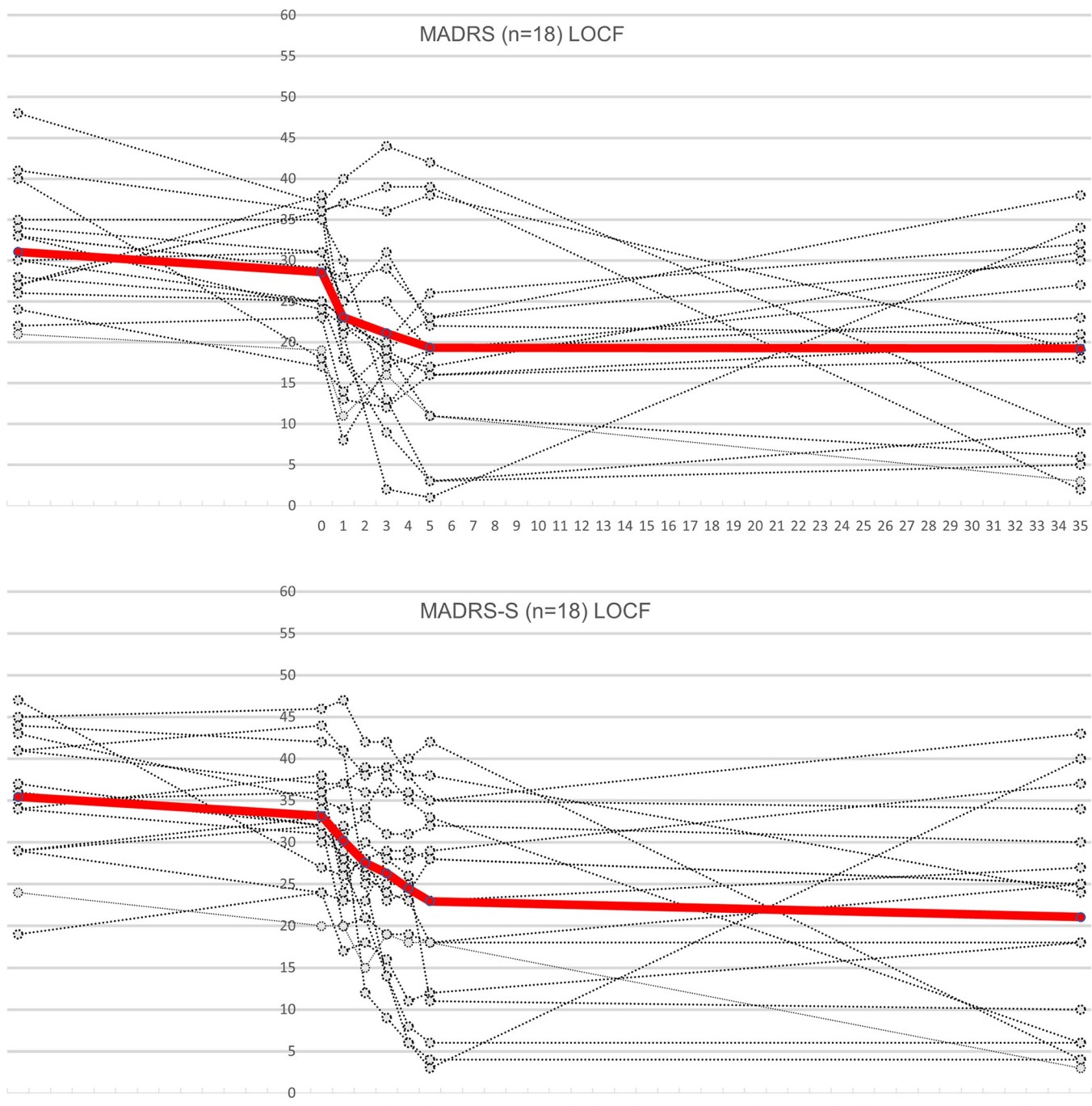

**Fig 2. Top: Individual Montgomery-Åsberg Depression Rating Scale (MADRS) scores in the PP-group.** Red line represents mean MADRS scores at all measured time-points (screening, 0 = baseline, 1 = after the first day of treatment, 3 = after the third day of treatment, 5 = after all treatments, 35 = at the 30-day follow-up). The three patients with the highest score (38,39 and 42) after day 5, all received ECT during the 30-day follow-up period. Two of those achieved lasting remission and one only met response criteria at follow-up. This patient had MADRS 12 after completing 5 ECT sessions but worsened some until follow-up. **Bottom: Individual Montgomery-Åsberg Depression Rating Scale Self-assessment (MADRS-S) scores in the PP-group.** Red line represents mean MADRS-S scores at all measured time-points (screening, 0 = baseline, 1 = after the first day of treatment, 2 = after the second day of treatment, 3 = after the third day of treatment, 4 = after the fourth day of treatment, 5 = after all treatments, 35 = at the 30-day follow-up).

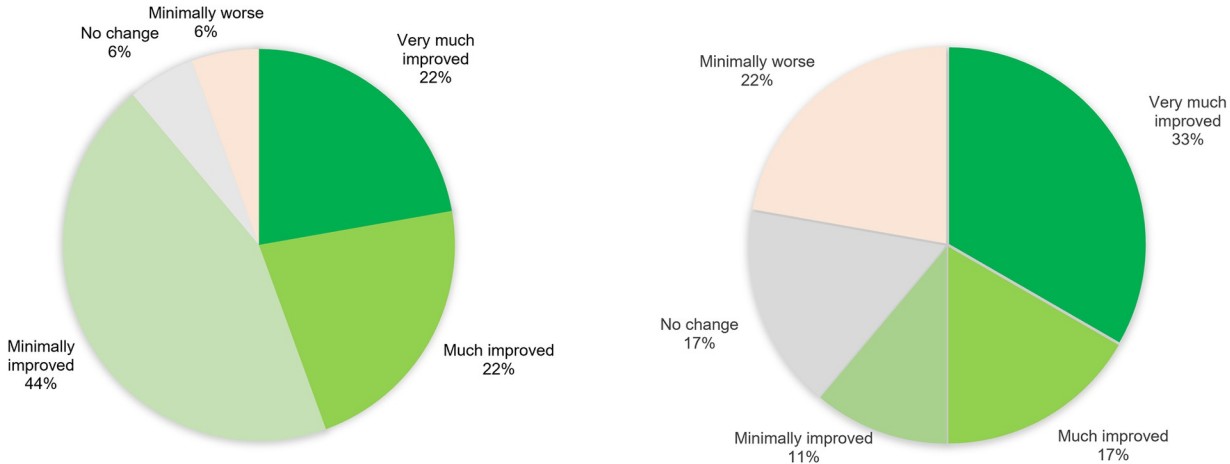

**Fig 3.** CGI-I after the last day of treatment (left) and at the 30 day follow-up (right) in the PP-group (n = 18). CGI-I Clinical Global Impression —Improvement. No patients were rated "much worse" or "very much worse". Three patients received ECT between the last day of treatment and the 30-day follow-up.

phase indicated high expectations regarding treatment outcomes, including hopes that accelerated treatment would reduce suffering and number of dropouts. Concerns were voiced regarding staffing, primarily related to a shortage of trained nurses and the project's vulnerability to

**Table 3.  Content analysis of pre-study nurse-interviews organized in categories, subcategories and examples of codes, representing relevant opinions expressed.**

| Category | Subcategory | Examples of codes |
|---|---|---|
| **High Expectations regarding treatment outcome** | Treatment could reach new patient groups | More in-patients. Exciting with primary care patients. Entire treatment as in-patient. |
| | Open for different outcomes | High expectations. Possible that acceleration makes a difference. Uncertain outcome. |
| | Effects of shorter treatment duration | Shortened suffering. Might be too intensive. Quicker than ECT. |
| | More efficient patient interaction | Less small talk. Better use of resources. Economical. |
| | Less potential side effects | Mild transient side effects. Possibly more dizziness. Could cause slightly more scalp pain. |
| | Increased retention rate | Fewer dropouts. Risk of patient fatigue. Probably motivated patients. |
| **Expectations regarding study participation** | Uncertainty of patient recruitment | Previous clinical recruitment was difficult. Find suitable patients. Attract different types of patients. |
| | Probably doable | No potential difficulties. More planning is necessary. Wise to try it out first |
| | Excited and confident | New and exciting. Not afraid of taking responsibility. Very motivated. |
| **Risk of premises being unsuitable and/or too small** | Overcrowding the main issue | More space needed. Will modify the recovery room. Multifunctional spaces. |
| | Cramped during ECT | Waiting patients disrupt workflow. Shared waiting room. Waiting patients is not disturbing. |
| | Confidentiality jeopardized | First names will be called out. Premises compromise integrity. Jeopardizes ECT-patients integrity. |
| | Unsuitable treatment room | Warm, badly ventilated treatment room. No windows. Long shifts are monotonous. |
| | Inadequate food logistics | Hope logistics work. Inadequate food logistics. |
| **Concerns about staffing** | Nurses need training | Too few trained nurses. Training takes time. Well staffed after training. |
| | Initially taxing for experienced staff | Difficult to start with. Workload evens out after training. Not unreasonable workload. |
| | Vulnerable to absence | Two nurses only work part time. TMS given in parallel with ECT. Doctors as backup in case of absence. |

**Table 4. Content analysis of post-study nurse-interviews organized in categories, subcategories and examples of codes, representing relevant opinions expressed.**

| Category | Subcategory | Examples |
|---|---|---|
| **Disappointing clinical outcomes** | Less effective than expected | Had hoped for better results. Fewer responders than expected. Same outcome as standard TMS. |
| | Manageable side effects | Tiredness and headaches. Fatigue could mask improvement. Got predicted side effects. |
| | Discouraging to patients | Some dropouts. Patients were motivated. Intense compared to sick leave. |
| **Study participation met staff expectations** | Participating was rewarding | Was instructive. Was interesting to participate. Everybody did their part. |
| | Concerns were well founded | My misgivings were confirmed. More difficult than expected. Turned out as expected. |
| **Premises unsuitable and too small** | Overcrowding | Expected crowding occurred. Better premises needed. Covid-19 complicated logistics. |
| | Cramped during ECT | Had to share spaces. Had to keep our voices down. Didn´t collide with ECT. |
| | Unsuitable treatment room | Work environment induced tiredness/headache. Bad air quality in the treatment room. Treatment room is too small. |
| | Unsuitable waiting room | Crowded in the mornings. Unable to offer a comfortable waiting room. Uncomfortable chairs. |
| **Undersized staffing** | Higher staffing level needed | Had to work alone. Too few nurses. Only two nurses work full time. |
| | Increased workload | Considerably higher workload. Work-intensity was too high. Same workload. |
| | Stressful for nurses | Demanding for nurses. Stressful for everyone. Stress decreased with practice. |
| | No time for administrative work | No time for administration. Time consuming rTMS-administration. Multiple treatments simultaneously. |
| **Worsened social work environment** | Communication issues | Problems with communication. Communication issues in the work-team. Nurses met each other less. |
| | Discontent among nurses | Worsened social work environment. Discontent among nurses. |
| **Constructive thoughts going forward** | Short treatment advantageous | Short treatment was good. Short treatment was smooth. |
| | Three patients/h too much | Two patients/h better. Three patients/h stressful. At least 20 minutes per patient needed. |
| | More resources needed | More resources needed. Good resource utilization. |
| | Need of a better protocol | Different protocols could be combined. Perhaps try four treatments/day for four weeks. |
| | Too early to evaluate | Too early to evaluate. |

their possible temporary absence. Additionally, nurses expressed reservations about the unit's premises regarding size and suitability for the project.

Subsequent interviews conducted after the study's completion confirmed concerns regarding inadequate facilities. Disappointment was prevalent concerning the perceived effectiveness of the treatment. The increased workload contributed to discontent within the workgroup, and also led to elevated stress levels, reduced time for administrative tasks, and more solitary work.

Nurse expectations regarding treatment effect were rated at 1.9 (min = 0, max = 3.5. SD = 1.4) pre-treatment. However, after completing the study, the nurse's assessment of treatment effect decreased to 0.8 (min = 0, max = 3. SD = 1.3). Expectations of side effects before the treatment phase were rated at -1 (min = -3, max = 1, SD = 1.6), which further decreased to -2 (min = -3, max = -1, SD = 1) after the treatment phase.

When queried about the practical aspects of the treatment using VAS, the patients in the PP-group rated it to 8.2 (min = 2, max = 10, SD = 2.2) after the first day of treatments. A slight reduction was noted after three days (mean = 7.7), but it returned to 8.2 after all treatments and at the 30-day follow-up (min = 2, max = 10, SD = 2.3).

## Discussion

The evaluation of new treatment protocols necessitates a careful assessment of their tolerability and safety profiles. Among the two serious adverse events (SAEs) reported, one involved a suicide attempt by a patient not responding to treatment, while experiencing severe depression with psychotic features—a condition often associated with less favorable outcomes in rTMS

treatment [18]. During follow-up, this patient achieved remission with ECT treatment, underscoring the need for close monitoring and access to other treatment options. The second SAE involved persistent headaches and renewed complaints of "buzzing in the head" in another nonresponder. Clinical assessment revealed no underlying neurological conditions. Although rTMS-induced tinnitus has been described in the literature [19], it was considered unlikely in this case, as this patient had a history of both auditory hallucinations and diverse psychosomatic symptoms spanning 20 years. A causal relationship between the reported symptoms and the study treatment was deemed very unlikely.

The 10% dropout rate matches data from the Swedish National Quality Register for Repetitive Transcranial Magnetic Stimulation (Q-rTMS) [8]. Both patients who dropped out did so due to severe anxiety, with one already reporting significant symptoms at baseline. The other patient eventually achieved remission three months after discontinuation, following standard iTBS treatment (30 sessions given over six weeks). While the dropout rate indicates similar tolerability for aTBS compared to standard iTBS, tailored support could prove valuable for managing anxiety and enhancing adherence.

Our rigorous approach assessing AEs continuously (34 evaluations per patient), captured a wide range of complaints, many reported only once by a single patient. Common AEs based on prior rTMS research were specifically queried, which might explain the higher prevalence of AEs compared to other studies [5, 7, 9], while providing a broader understanding of aTBS tolerability. Most AEs were transient, as shown by 30-day follow-up data, where few patients reported AEs, and VAS side effect ratings were near 10 ("no side effects at all"). We found no cognitive side effects; instead, there was a reported improvement in memory functioning, possibly reflecting a short-term direct cognitive enhancing effect of HF-rTMS [20] and/or an improvement in depressive state.

We observed a substantial reduction of depressive symptoms at the group level. However, the reduction in MADRS-S-score and the number of patients achieving remission and response were more in line with the Q–rTMS data [4, 8] than the rates reported in the SAINT/SNT studies [9, 21]. This finding coincides with the emerging evidence for faster response, but similar efficacy from accelerated protocols compared to standard treatment [22]. The SAINT/SNT protocol conspicuously used individualized fcMRI-guided treatment target identification, although the importance of this has been questioned [23]. The trend towards better effect with more treatments per day, more treatments in total and more total pulses, might further explain the observed difference in results. However, isolating the impact of each variable remains inherently difficult [22].

Another important difference from Cole et al.'s studies concern patient selection. While our patients were recruited from community-based clinics, participants in the SAINT study were also recruited through a Depression Research Clinic at Stanford University and study advertisements. Although our cohort had a higher age of depression onset and lower MADRS-scores at baseline, on average they had been hospitalized twice for depression and almost half of them had prior experience with ECT, possibly reflecting their overall severity of illness. In contrast no participant in the SAINT study had undergone ECT.

The optimal stimulation intensity for aTBS-protocols is not presently known. In a recent pilot study exploring the safety and efficacy of a pragmatic aTBS-protocol for treatment resistant depression [24], 36 iTBS-sessions were delivered over five consecutive days with 600 pulses per session and the average intensity of 93.98% (SD 16.19) of the resting motor threshold. High and sustained rates of response (70%) and remission (55%) were reported four weeks post treatment. The SAINT/SNT protocol also used a lower resting motor threshold adjusted to 90% of the resting motor threshold at the treatment target. This difference in treatment parameters may partially explain the variation in outcomes observed between the

protocols, as suggested by a recent review [25] proposing that sub-motor stimulation intensities could be more effective for TBS protocols including aTBS.

SATIS follow-up depression scores were higher than post-intervention scores which differs from the pattern of ongoing improvement observed in a meta-analysis of accelerated rTMS protocols [26]. This pattern was however not observed with aTBS protocols, consistent with our data. The relapse rates in our cohort could also be attributed to factors like level of treatment resistance, diagnostic heterogeneity and comorbidity load.

While some patients demonstrated the typical rapid response observed in other aTBS-studies [9, 21, 24], the low positive correlation between higher pretreatment expectations and greater improvements in depression symptoms suggest that patient expectations may influence treatment outcomes. Given the small sample size, these findings should be interpreted with caution, but they support the potential benefit of addressing patient expectations in depression treatment strategies and contributes to the ongoing efforts to untangle the placebo-effect in rTMS treatment [27].

We also explored the feasibility of implementing aTBS in a clinical setting based on the framework for evaluating complex interventions from the The UK Medical Research Council [28]. This framework emphasizes the importance of recognizing the interaction between the intervention and its context. By including stakeholder perspectives, we gained valuable insights into the gap between nurse expectations and perceived treatment effectiveness. This underscores the need for realistic expectations when adopting innovative treatments. The increased workload and stress experienced by nursing staff, particularly from 8 AM to 10 AM, when three of them were dedicated to administering ECT, highlight the importance of proper staffing and support to ensure successful implementation. Admitting patients to a day clinic could resolve the logistical challenges between treatment sessions. Patient ratings using VAS indicated few practical problems with the protocol during and after treatment. Future feasibility studies should include comprehensive interviews with patients to gain a deeper understanding of their experience, identifying specific areas for improvement in the treatment protocol and support systems.

Given the capacity of a single rTMS-equipment and seven hours of nurse-time per day, the theoretical number of patients that can be treated in four weeks is lower using aTBS (12) compared to standard iTBS (21), increasing per-patient cost by 50%. This reduced throughput requires extended operating hours or additional equipment to treat multiple patients simultaneously, adversely impacting budget further.

Decisionmakers must weigh these pronounced logistical and economic challenges against the possible clinical benefits, such as faster response time, when considering accelerated treatment.

### Limitations

The absence of a control group and the small sample size limit the ability to establish causality or generalize the findings. Additionally, the short follow-up period does not capture the durability of aTBS treatment. Another important limitation is the heterogeneity of the population. Unlike many TMS-trials, which often exclude participants with high suicide risk, psychotic features, bipolar depression or co-morbid conditions like autism, our study included a more diverse cohort. This approach was intended to reflect real-world patients and to facilitate comparison with Q-rTMS. However, it also introduces variability that may obscure treatment effect found in more narrowly defined groups.

### Conclusion

The SATIS study offers valuable lessons on aTBS as a clinically implemented treatment for depression. While the results suggest preliminary effectiveness, the 17% remission rate raises

questions about how aTBS compares to established interventions like ECT, which shows remission rates over 50% in similar populations [29, 30]. Therefore, future research should not only focus on optimizing aTBS protocols for efficacy, tolerability, and safety, but also directly compare its effectiveness against other treatments. Larger samples and longer follow-ups, including assessments of functional and quality of life outcomes, are crucial for integrating aTBS into the therapeutic landscape for depression. SATIS marks an important step in understanding aTBS, providing a foundation for future clinical research.

## Supporting information

**S1 Table. Psychotropic drugs regularly taken during the five-day treatment phase of SATIS.**
(PDF)

**S2 Table. Other reported adverse events (AEs) at any time (n = 20).** AEs reported by only one patient: Toothache, right arm paresthesia, transient hypersomnia, light-headedness, jaw discomfort during mastication, panic attack, supraorbital pain, tremor of the hands, slight spatial disorientation, bilateral shoulder myalgia, warm sensation under the coil, vertigo during treatment, buzzing in the head, feeling dazed, anticipatory anxiety, tension in the head, renal colic.
(PDF)

**S1 Fig. Top: Depression score before and after treatment in the PP-group (n = 18).** Whiskers min—max, box median, 1st and 3rd quartile. X marks the mean. **Bottom: Depression score before and after treatment in the ITT group (n = 20).** Whiskers min—max, box median, 1st and 3rd quartile. X marks the mean.
(TIF)

**S2 Fig. Depression score before and after treatment and at 30-day follow-up, in the PP-group (n = 15) that finished the treatment program and did not receive ECT-treatment between finishing the aTBS treatment and the 30-day follow-up.** Whiskers min—max, box median, 1st and 3rd quartile. X marks the mean.
(TIF)

**S3 Fig. Individual and mean EQ-5D VAS scores in the PP-group (0 = baseline, 1 = after the first day of treatment, 2 = after the second day of treatment, 3 = after the third day of treatment, 4 = after the fourth day of treatment, 5 = after all treatments, 35 = at the 30-day follow-up).**
(TIF)

**S4 Fig. Correlation scatter plot showing the relation between change in depression score from before to after treatment to self-rated expectations at baseline.** The three outliers represent patients with a very robust treatment response—a reduction in Montgomery-Åsberg depression rating scale (MADRS)-score of around 30 points.
(TIF)

**S1 Data. Excel data file with all collected data.**
(XLSX)

## Acknowledgments

We would like to thank the registered nurses Johanna Karlström, Paola Vera, Catrine Härdfeldt-Olsson, Gunilla Unosson and Linda Rosén for their enthusiastic participation in the

study as well as healthcare documentation specialist Susanne Nylander for transcribing the nurse interviews.

Declaration of generative AI and AI-assisted technologies in the writing process:

During the preparation of this work the authors used ChatGPT 4 to improve language in the discussion section. After using this service, the authors reviewed and edited the content as needed and take full responsibility for the content of the publication.

## Author Contributions

**Conceptualization:** Marcus Persson, Viktor Fabri, Alexander Reijbrandt, Pouya Movahed Rad.

**Formal analysis:** Marcus Persson, Viktor Fabri.

**Funding acquisition:** Marcus Persson, Pouya Movahed Rad.

**Investigation:** Marcus Persson, Viktor Fabri, Alexander Reijbrandt, Hans Eriksson.

**Methodology:** Marcus Persson, Viktor Fabri, Alexander Reijbrandt, Hans Eriksson, Pouya Movahed Rad.

**Resources:** Pouya Movahed Rad.

**Supervision:** Annika Lexén, Pouya Movahed Rad.

**Writing – original draft:** Marcus Persson, Viktor Fabri.

**Writing – review & editing:** Annika Lexén, Hans Eriksson, Pouya Movahed Rad.

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
