## [Decision Letter · Decision Letter 0]

22 Aug 2024

PONE-D-24-25365The Scania Accelerated Intermittent Theta-burst Implementation Study (SATIS) – lessons from an accelerated treatment protocolPLOS ONE

Dear Dr. Persson,

Thank you for submitting your manuscript to PLOS ONE. After careful consideration, we feel that it has merit but does not fully meet PLOS ONE’s publication criteria as it currently stands. Therefore, we invite you to submit a revised version of the manuscript that addresses the points raised during the review process.

We look forward to receiving your revised manuscript.

Kind regards,

Mu-Hong Chen, M.D., Ph.D.

Academic Editor

PLOS ONE

Journal Requirements:

2. Thank you for stating the following in the Competing Interests section: "I have read the journal´s policy and the authors of this manuscript have the following competing interests: Authors MP, VF, AR, AL, and PM none, HE is a full-time employee at HMNC Brain Health, a pharmaceutical company developing pharmaceutical treatments for depression."

5. We notice that your supplementary figures are included in the manuscript file. Please remove them and upload them with the file type 'Supporting Information'. Please ensure that each Supporting Information file has a legend listed in the manuscript after the references list.

Reviewers' comments:

Reviewer's Responses to Questions

**Comments to the Author**

1. Is the manuscript technically sound, and do the data support the conclusions?

Reviewer #1: Yes

Reviewer #2: Yes

2. Has the statistical analysis been performed appropriately and rigorously? 

Reviewer #1: Yes

Reviewer #2: Yes

3. Have the authors made all data underlying the findings in their manuscript fully available?

Reviewer #1: Yes

Reviewer #2: Yes

4. Is the manuscript presented in an intelligible fashion and written in standard English?

Reviewer #1: Yes

Reviewer #2: Yes

5. Review Comments to the Author

Reviewer #1: 1. In common TMS trials, participants with a high risk of suicide are often excluded, as well as those with psychotic features. Due to concerns about heterogeneity, cases with bipolar depression may also be excluded, or the trials may specifically focus on including only those with bipolar depression. Additionally, it is relatively rare to include cases with comorbid autism. Many trials also focus on treatment-resistant depression (TRD), most commonly defined as inadequate response to two antidepressants. Overall, these trials are usually designed this way to enhance the homogeneity of the participants. In this study, despite the small sample size and the high heterogeneity of the population, it is acceptable that the TMS outcomes are closer to those observed in national registry data rather than in a well-designed SAINT protocol study. This is actually more reflective of real-world clinical practice with TMS. However, the author should mention the limitations regarding the participants.

2. In addition, please describe the participants' medication usage status.

3. I would like to know the rationale behind designing the study with 5-7-7-7-7 sessions. Why not have 7 sessions each day for 5 days or 5 sessions each day? I suspect the author aimed to reduce the total sessions while increasing the MT% to compare with the SAINT Protocol. However, this should be explained in the Introduction.

4. Line 360 mentions that SAINT, with its higher number of sessions and total pulses, might be related to better outcomes. There is a considerable amount of research on TMS total pulses/total sessions and treatment effects (though not necessarily in the context of accelerated protocols). The author should delve deeper into this aspect—does more frequent stimulation necessarily lead to better effectiveness?

(PMID: 38183740, PMID: 38723735)

5. In the same vein, Line 373, The author mentions that the SAINT protocol uses a lower MT%, which is associated with better outcomes (??). This aspect also requires a more thorough discussion.

6. It is generally believed that the placebo effect can be quite significant in rTMS research (PMID: 29111404). In this study, many cases showed an immediate response within 2-3 days. Has the possibility of the placebo effect been considered?

Reviewer #2: I appreciate the invitation to review this work. The manuscript is intriguing, current, significant, and well-written. It explores the effectiveness of SATIS protocols, a modified version of SAINT protocols, and provides a practical feasibility analysis in a clinical setting, addressing the challenges of medical economics. Interestingly, the effectiveness of SATIS was shown to be comparable or maybe lower than standard iTBS protocols, but not comparable to the SAINT technique. I have a few minor comments to share:

1. 1. When comparing SATIS protocols to general accelerated TMS protocols (which consist of 20-30 sessions administered over a short period), SATIS protocols were found to be more similar to SAINT procedures than the general aTMS protocols. Consequently, readers may be interested in understanding why the results deviated significantly from the SAINT procedures. Therefore. The current discussion may be not enough. I believe that a more comprehensive explanation should be provided regarding the discrepancy observed in the results of the SAINT protocol, as well as the potential underlying confounding factors. For instance, it would be beneficial to include further discussions regarding the overall number of sessions (33 compared to 50 sessions), the motor threshold (subthreshold versus suprathreshold), the localization methods, and the features of the patients. It is also important to include the relevant citation. For instance, the reference should be added for LN364-367 that suggests that the refractoriness of individuals, as shown by their experience of electroconvulsive therapy (ECT) and hospitalization, could potentially impact the success of SATIS.

2. Furthermore, it is suggested to perform additional analysis and outcomes using the same exclusion criteria and remission definition in order to compare the results with SAINT. The SAINT study enrolled patients diagnosed with major depressive disorder only while excluding individuals with bipolar I disorder, bipolar II disorder, major depressive disorder with psychotic features, and autism spectrum disorder. The impact of rTMS on individuals’ comorbid with those diagnoses remains unknown. Furthermore, they identified remitters if they met these specific criteria at any point during the 4-week follow-up.

[Am J Psychiatry . 2022 Feb;179(2):132-141. doi: 10.1176/appi.ajp.2021.20101429. Epub 2021 Oct 29. Stanford Neuromodulation Therapy (SNT): A Double-Blind Randomized Controlled Trial]

3. The inclusion criteria was patients having baseline MADRS≧20. However, LN245 and LN248 reported the min MADRS was 17 at baseline no matter in PP group or ITT group. Please explain the reason.

4. The readers may be interested in knowing the average percentage changes in depression scores before and after the SATIS protocol and the results of pair-t test before and after SATIS treatment.

5. LN376-378 “SATIS follow-up depression scores were higher than post-intervention scores. Interestingly a meta-analysis suggested ongoing improvement post-accelerated rTMS treatment, a pattern not observed with aTBS, consistent with our data.” The paragraph is unclear. It is uncertain whether the SATIS results align with the findings in the literature or not. Furthermore, if there is no compatibility, could you provide a rationale for the "relapsed" depression ratings observed throughout the follow-up period?

6. Table 1 provides information on the number of participants who had undergone Electroconvulsive Therapy (ECT) at least once in their lifetime (n=8). Is it feasible to determine the response or remission rate of earlier electroconvulsive therapy (ECT) in those patients? If the patients did not respond to earlier electroconvulsive therapy (ECT) treatment, particularly after receiving more than six to eight sessions of ECT, it would be more reasonable to expect them to not respond to the SATIS regimen.

6. PLOS authors have the option to publish the peer review history of their article (what does this mean?). If published, this will include your full peer review and any attached files.

Reviewer #1: No

Reviewer #2: **Yes: **Chih-Ming Cheng

---

## [Author Response · Author response to Decision Letter 0]

17 Sep 2024

Response to reviewers

Comments from the academic editor:

1. Style requirements were double checked.

2. We have updated the Competing interests statement in our new cover letter. 

3a. After a discussion amongst the authors we have concluded that it is in accordance with our ethical approval to share all data. We were concerned that sharing specific data (ie.age and sex) would have posed a risk of inadvertently reveal the identities of individual participants, as only a couple of hundred patients in our region have been treated with rTMS, and only those in this study (20 individuals) received an accelerated protocol. However, we realize that for instance sex is a relevant parameter to consider if you want to include our data in a future meta-analysis. There are no restrictions imposed by a third-party organization or any other body.

b. We have decided to include our data as supporting information files. Our Data availability statement has been updated. 

4. Our ethics statement only appears in the Methods section of our manuscript.

5. We put supporting information tables and legends after the references in the manuscript file. 

6. And added captions. 

Comments from the reviewers:

Reviewer #1: 

1. In common TMS trials, participants with a high risk of suicide are often excluded, as well as those with psychotic features. Due to concerns about heterogeneity, cases with bipolar depression may also be excluded, or the trials may specifically focus on including only those with bipolar depression. Additionally, it is relatively rare to include cases with comorbid autism. Many trials also focus on treatment-resistant depression (TRD), most commonly defined as inadequate response to two antidepressants. Overall, these trials are usually designed this way to enhance the homogeneity of the participants. In this study, despite the small sample size and the high heterogeneity of the population, it is acceptable that the TMS outcomes are closer to those observed in national registry data rather than in a well-designed SAINT protocol study. This is actually more reflective of real-world clinical practice with TMS. However, the author should mention the limitations regarding the participants.

Thank you for your insightful comments. We have now addressed this point in the manuscript under the "Limitations" section.

2. In addition, please describe the participants' medication usage status. 

Thank you for your comment. In response, we have added a comprehensive table (S1) to the supporting information, which details all psychotropic drugs regularly taken by each participant during the treatment phase of the study. 

3. I would like to know the rationale behind designing the study with 5-7-7-7-7 sessions. Why not have 7 sessions each day for 5 days or 5 sessions each day? I suspect the author aimed to reduce the total sessions while increasing the MT% to compare with the SAINT Protocol. However, this should be explained in the Introduction. 

Thank you for your comment. The decision to administer only five treatments on the first day was driven solely by practical considerations. At the start of treatment, considerably more time is needed to prepare the patient with necessary information, set up the equipment, determine the individual motor threshold, and identify and calibrate the treatment target. We have revised the manuscript accordingly. Ideally, we would have administered 10 sessions per day, in line with the SAINT protocol, but this was not feasible due to the working hours of our personnel. The SATIS protocol was the most intensive approach we could implement under these constraints. Since we did not have the means to measure the depth-corrected motor threshold (MT) percentage, we decided to use the same MT as the standard iTBS protocol, which is already widely established in Sweden. Notably, the MT in the SAINT protocol was also set to a maximum of 120%. 

4. Line 360 mentions that SAINT, with its higher number of sessions and total pulses, might be related to better outcomes. There is a considerable amount of research on TMS total pulses/total sessions and treatment effects (though not necessarily in the context of accelerated protocols). The author should delve deeper into this aspect—does more frequent stimulation necessarily lead to better effectiveness? (PMID: 38183740, PMID: 38723735)

Thank you for your comment and for suggesting these relevant articles. While we recognize the importance of discussing the specific factor of total pulses/total sessions, we chose not to include the studies by Hsu et al. and Yu et al. despite being aware of their research. This decision was made because these studies as you mention use data from protocols that do not involve accelerated or theta burst protocols. We understand and agree that the underlying question you raise, whether more pulses necessarily lead to better outcomes, is crucial and worth discussing. We have now addressed this issue in the manuscript, although we have opted not to include the suggested references.

5. In the same vein, Line 373, The author mentions that the SAINT protocol uses a lower MT%, which is associated with better outcomes (??). This aspect also requires a more thorough discussion.

Thank you for your comment on the motor threshold percentage (MT%) and its impact on treatment outcomes. We have now expanded the discussion in the manuscript.

6. It is generally believed that the placebo effect can be quite significant in rTMS research (PMID: 29111404). In this study, many cases showed an immediate response within 2-3 days. Has the possibility of the placebo effect been considered? 

Thank you for your valuable feedback. We have expanded the discussion and limitations section to acknowledge the importance of the placebo effect and effective blinding in rTMS research, referencing the relevant literature (PMID: 29111404). Although this is an important consideration, our study focused on feasibility and did not include a control group to assess efficacy. The absence of a control group limits our ability to establish causality or generalize the findings. Future studies with blinded controls are needed to fully evaluate the placebo effect, as highlighted by (PMID: 29111404).

Reviewer #2: 

I appreciate the invitation to review this work. The manuscript is intriguing, current, significant, and well-written. It explores the effectiveness of SATIS protocols, a modified version of SAINT protocols, and provides a practical feasibility analysis in a clinical setting, addressing the challenges of medical economics. Interestingly, the effectiveness of SATIS was shown to be comparable or maybe lower than standard iTBS protocols, but not comparable to the SAINT technique. I have a few minor comments to share:

1. When comparing SATIS protocols to general accelerated TMS protocols (which consist of 20-30 sessions administered over a short period), SATIS protocols were found to be more similar to SAINT procedures than the general aTMS protocols. Consequently, readers may be interested in understanding why the results deviated significantly from the SAINT procedures. Therefore. The current discussion may be not enough. I believe that a more comprehensive explanation should be provided regarding the discrepancy observed in the results of the SAINT protocol, as well as the potential underlying confounding factors. For instance, it would be beneficial to include further discussions regarding the overall number of sessions (33 compared to 50 sessions), the motor threshold (subthreshold versus suprathreshold), the localization methods, and the features of the patients. It is also important to include the relevant citation. For instance, the reference should be added for LN364-367 that suggests that the refractoriness of individuals, as shown by their experience of electroconvulsive therapy (ECT) and hospitalization, could potentially impact the success of SATIS.

Thank you for raising these important concerns. Since these issues overlap significantly with concerns raised by Reviewer #1, they have already been addressed accordingly. Specifically, the discussion now includes a more detailed comparison between the SATIS and SAINT protocols, addressing the differences in the number of sessions (33 vs. 50), stimulation intensity (suprathreshold vs. subthreshold), localization methods (MRI-guided vs. fcMRI-guided), and patient characteristics such as refractoriness, prior experience with ECT, and hospitalization history. These points are elaborated in the discussion section, providing explanations for the discrepancies in outcomes between SATIS and SAINT, supported by relevant citations.

2. Furthermore, it is suggested to perform additional analysis and outcomes using the same exclusion criteria and remission definition in order to compare the results with SAINT. The SAINT study enrolled patients diagnosed with major depressive disorder only while excluding individuals with bipolar I disorder, bipolar II disorder, major depressive disorder with psychotic features, and autism spectrum disorder. The impact of rTMS on individuals’ comorbid with those diagnoses remains unknown. Furthermore, they identified remitters if they met these specific criteria at any point during the 4-week follow-up. [Am J Psychiatry . 2022 Feb;179(2):132-141. doi: 10.1176/appi.ajp.2021.20101429. Epub 2021 Oct 29. Stanford Neuromodulation Therapy (SNT): A Double-Blind Randomized Controlled Trial]

Thank you for your suggestion regarding additional analyses with exclusion criteria and remission definitions aligned with the SAINT study. We have carefully considered this approach. However, given the already small sample size in our study, we believe that a post hoc subgroup analysis may not yield informative or generalizable results. Additionally, the primary objective of our study was not to assess efficacy but rather to evaluate the feasibility of implementing this protocol in routine clinical practice. We acknowledge the importance of addressing cohort heterogeneity, and have expanded the discussion on this issue in the revised manuscript. We hope this clarifies our position and appreciate your understanding.

3. The inclusion criteria was patients having baseline MADRS≧20. However, LN245 and LN248 reported the min MADRS was 17 at baseline no matter in PP group or ITT group. Please explain the reason. 

Thank you for pointing this out. This discrepancy is due to the fact that three patients experienced a spontaneous decrease in their MADRS scores between inclusion and the baseline measurement at the start of treatment. This has been clarified in the manuscript under the results section.

4. The readers may be interested in knowing the average percentage changes in depression scores before and after the SATIS protocol and the results of pair-t test before and after SATIS treatment.

Thank you for your suggestion regarding the inclusion of additional statistical analyses. We have now included the average percentage changes in depression scores before and after the SATIS protocol, as well as the results of a paired t-test, in the results section of the manuscript.

5. LN376-378 “SATIS follow-up depression scores were higher than post-intervention scores. Interestingly a meta-analysis suggested ongoing improvement post-accelerated rTMS treatment, a pattern not observed with aTBS, consistent with our data.” The paragraph is unclear. It is uncertain whether the SATIS results align with the findings in the literature or not. Furthermore, if there is no compatibility, could you provide a rationale for the "relapsed" depression ratings observed throughout the follow-up period?

Thank you for pointing out the need for clarity in this section. We have revised the paragraph to better explain how the SATIS results compare with findings in the literature. We have also provided a rationale for the observed "relapsed" depression ratings during the follow-up period, considering the specific characteristics of our patient population and treatment protocol. We hope this addresses your concerns.

6. Table 1 provides information on the number of participants who had undergone Electroconvulsive Therapy (ECT) at least once in their lifetime (n=8). Is it feasible to determine the response or remission rate of earlier electroconvulsive therapy (ECT) in those patients? If the patients did not respond to earlier electroconvulsive therapy (ECT) treatment, particularly after receiving more than six to eight sessions of ECT, it would be more reasonable to expect them to not respond to the SATIS regimen.

Thank you for your comment. As we mentioned in response to comment nr 2, we believe that our small sample size does not support subgroup analyses in a way that would yield generalizable and meaningful information. However, after conducting a post-hoc analysis of our data, we found the following:

Seven patients who underwent treatment had previously received ECT, though not recently. These patients showed an average reduction of 9.0 points in MADRS. Of these, 1 achieved remission and six were non-responders.

Eleven patients who underwent treatment had not received ECT previously. These patients showed an average reduction of 9.4 points in MADRS. Two patient achieved remission, 3 were responders while eight were non-responders.

We believe that this subgroup analysis does not add significant value, considering that it is a post-hoc analysis with a small sample size.

---

## [Decision Letter · Decision Letter 1]

10 Dec 2024

The Scania Accelerated Intermittent Theta-burst Implementation Study (SATIS) – lessons from an accelerated treatment protocol

PONE-D-24-25365R1

Dear Dr. Marcus Persson,

We’re pleased to inform you that your manuscript has been judged scientifically suitable for publication and will be formally accepted for publication once it meets all outstanding technical requirements.

Kind regards,

Mu-Hong Chen, M.D., Ph.D.

Academic Editor

PLOS ONE

Additional Editor Comments (optional):

Reviewers' comments:

Reviewer's Responses to Questions

**Comments to the Author**

1. If the authors have adequately addressed your comments raised in a previous round of review and you feel that this manuscript is now acceptable for publication, you may indicate that here to bypass the “Comments to the Author” section, enter your conflict of interest statement in the “Confidential to Editor” section, and submit your "Accept" recommendation.

Reviewer #1: All comments have been addressed

Reviewer #2: All comments have been addressed

2. Is the manuscript technically sound, and do the data support the conclusions?

Reviewer #1: Yes

Reviewer #2: Yes

3. Has the statistical analysis been performed appropriately and rigorously? 

Reviewer #1: Yes

Reviewer #2: Yes

4. Have the authors made all data underlying the findings in their manuscript fully available?

Reviewer #1: Yes

Reviewer #2: Yes

5. Is the manuscript presented in an intelligible fashion and written in standard English?

Reviewer #1: Yes

Reviewer #2: Yes

6. Review Comments to the Author

Reviewer #1: I have no further comment. Despite the high heterogeneity of the population and the limited sample size, it remains a study worth publishing.

Reviewer #2: (No Response)

7. PLOS authors have the option to publish the peer review history of their article (what does this mean?). If published, this will include your full peer review and any attached files.

Reviewer #1: No

Reviewer #2: No

---

## [Editor Report · Acceptance letter]

18 Dec 2024

PONE-D-24-25365R1 

PLOS ONE

Dear Dr. Persson, 

I'm pleased to inform you that your manuscript has been deemed suitable for publication in PLOS ONE. Congratulations! Your manuscript is now being handed over to our production team.

Kind regards, 

on behalf of

Dr. Mu-Hong Chen 

Academic Editor

PLOS ONE